# Agricultural Use of Copper and Its Link to Alzheimer’s Disease

**DOI:** 10.3390/biom10060897

**Published:** 2020-06-12

**Authors:** Fábio C. Coelho, Rosanna Squitti, Mariacarla Ventriglia, Giselle Cerchiaro, João P. Daher, Jaídson G. Rocha, Mauro C. A. Rongioletti, Anna-Camilla Moonen

**Affiliations:** 1Phytotechnics Laboratory, Universidade Estadual do Norte Fluminense Darcy Ribeiro—UENF; Campos dos Goytacazes, RJ 28013-602, Brazil; jaidsongr@yahoo.com.br; 2Molecular Markers Laboratory, IRCCS Instituto Centro San Giovanni di Dio Fatebenefrate lli, 25125 Brescia, Italy; 3Fatebenefratelli Foundation for Health Research and Education, AFaR Division, 00186 Rome, Italy; mariacarla.ventriglia@afar.it; 4Center for Natural Science and Humanities, Federal University of ABC—UFABC, Santo André, SP 09210-580, Brazil; gicerchiaro@gmail.com; 5Hospital Universitário Antônio Pedro, Universidade Federal Fluminense, Niterói, RJ 24210-350, Brazil; jpldaher@gmail.com; 6Department of Laboratory Medicine, Research and Development Division, San Giovanni Calibita Fatebenefratelli Hospital, Isola Tiberina, 00186 Rome, Italy; biomolfbf@gmail.com; 7Land Lab, Institute of Life Sciences, Scuola Superiore Sant’Anna, 56127 Pisa, Italy; c.moonen@santannapisa.it

**Keywords:** heavy metal, dementia, organic agriculture, agroecology

## Abstract

Copper is an essential nutrient for plants, animals, and humans because it is an indispensable component of several essential proteins and either lack or excess are harmful to human health. Recent studies revealed that the breakdown of the regulation of copper homeostasis could be associated with Alzheimer’s disease (AD), the most common form of dementia. Copper accumulation occurs in human aging and is thought to increase the risk of AD for individuals with a susceptibility to copper exposure. This review reports that one of the leading causes of copper accumulation in the environment and the human food chain is its use in agriculture as a plant protection product against numerous diseases, especially in organic production. In the past two decades, some countries and the EU have invested in research to reduce the reliance on copper. However, no single alternative able to replace copper has been identified. We suggest that agroecological approaches are urgently needed to design crop protection strategies based on the complementary actions of the wide variety of crop protection tools for disease control.

## 1. Alzheimer’s Disease

Alzheimer’s disease (AD) is the most common cause of dementia in the elderly. The World Health Organization (WHO) [1] estimated the ranking of the top seven countries in the number of deaths caused by AD. In absolute numbers, the ranking was the China, USA; India, United Kingdom, Indonesia, France and Germany with 563.5; 259.5; 140.9; 82.5; 54.7, 49.6 and 48.8 thousand deaths in 2016, respectively.

The late-onset sporadic form of the disease appears after age 65, accounts for 90–95% of AD cases and derives from a complex interaction of genetic and non-genetic factors. The early-onset familial AD represents a small portion of all AD cases and is due to mutations in *APP, PSEN1* and *PSEN2* genes. The disease is caused by misfolding of specific proteins that are associated with extracellular deposits of the beta-amyloid (Aβ) protein [2,3] and aberrant aggregation of pathological forms of the intraneuronal protein Tau [4].

Alzheimer’s disease accounts for 60–80% of cases of dementia and autopsy studies indicate that 50% involve solely AD pathology [5]. Aβ plaques may contribute to neurodegeneration, but the etiology of the disease is still unclear and several hypotheses have been posited. In the 1990s, the ‘amyloid cascade’ hypothesis became the dominant hypothesis. A consistent number of clinical trials focusing on Aβ cleaving enzymes, antibodies or anti-Aβ compounds so far, have provided negative results [6]. However, a recent re-evaluation (in December 2019) of results from a phase III clinical trial with aducanumab (EMERGE study), appears encouraging, suggesting that aducanumab can both remove Aβ plaques and slow cognitive decline in people with early AD. Surprisingly in the same month, China approved for AD treatment Oligomannate, and a phase III study was planned for starting in 2020 in China, the United States and Europe to authorize the marketing of the drug [7].

Besides these encouraging results that have still to be fully assessed, recent studies, pointing to prevention strategies, showed the predictive accuracy of the ‘LIfestyle for BRAin Health’ (LIBRA) score for dementia and mild cognitive impairment (MCI), a prodromal stage of the disease [8,9,10,11]. In this line, lifestyle intervention [12] has been shown to benefit cognition, as demonstrated by results from the FINGER [13], and the Rotterdam studies [14]. As a whole, the latest evidence points out to the multifactorial etiology of AD and the potential of preventive strategies to reduce the prevalence of the disease.

## 2. Alzheimer’s Disease and Copper

Copper is an essential trace metal controlling human physiology. Brain cells use copper during development, and it is indispensable in vital processes such as respiration, energy production, formation of myelin sheath around neurons, synthesis of neurotransmitters, immune system responses, collagen and pigment synthesis and wound healings. We surveyed the number of papers in the medicine database on the biological connection between AD and copper. The search on ‘Scopus’ with the terms “Alzheimer’s disease and copper” provided 3009 document results (Figure 1).

The stratification by ‘subject area’ of the Scientific Journals of publication can help depict the large areas of scientific articles published on this topic that enclose.

### 2.1. Copper Connection with Alzheimer’s Disease: Biochemistry Literature

This literature includes Inorganic and Coordination Chemistry and is mostly focused on the interaction of Aβ and the Aβ precursor protein (APP, encoded by the *APP* gene) with copper. APP is a copper protein [15]. Consolidate knowledge indicates that the APP/Aβ system is central for AD pathogenesis, and a recent view proposes that the APP/Aβ system is centrally involved in neuronal copper transport at the synapses and in processes of learning and memory [16,17,18]. Loosely bound copper, as a transition metal, actively facilitates oxidative stress via Fenton like and Haber Weiss reactions. These reactions have been demonstrated to result in Aβ oligomer formation and their precipitation within plaques along with lipid peroxidation [19,20,21,22]. Established evidence demonstrated that: APP is a copper protein that binds and reduces copper from Cu(II) to Cu(I) [15], facilitating copper-induced toxicity in cell cultures and oxidative stress through the production of H_2_O_2_ [23]; that Aβ and metals are packed together in the brain Aβ plaques [15,24,25]; that Aβ plaques can be dissolved by chelating agents, which sequester the copper [24]. Overload of Cu(II)-Aβ molecules are also probable, particularly at the synapses where both are released [16]: copper ions are released from synaptic vesicles reaching concentrations as high as 15 μmol/L in a form of labile copper, not bound to proteins [26]. These conditions facilitate Cu-Aβ formation. At the same time, cell-associated copper into neurons could be decreased [27].

### 2.2. Copper Connection with Alzheimer’s Disease: Medicine Literature

Most of these studies focused on the comparison of copper levels in diverse organ tissues or biological matrices (e.g., serum, plasma, cerebrospinal fluid, brain, hair, nails). Most of these findings have been evaluated through meta-analyses that reported excess copper in general circulation [28,29,30] and copper deficiency in the brain [31]. The picture provided supports the evidence that patients with AD fail to maintain a correct metabolic balance and distribution of copper in the body [18]. In line with these findings, a recent meta-analysis has attributed the copper excess found in AD serum to the expansion of the fraction of exchangeable Cu^2+^ defined as copper not bound to ceruloplasmin (non-ceruloplasmin copper, also referred to as ‘free’ copper) [29]. This serum fraction is composed of copper loosely bound to albumin, α2 macroglobulin, peptides and amino acids and exchanged among them [29]. Non-ceruloplasmin copper has been already identified as a marker of Wilson disease, a rare inborn error of copper metabolism and a paradigm of non-ceruloplasmin copper toxicosis and accumulation [18]. Overall, existing meta-analyses in AD provided results of decreased levels of copper in the brain [31], along with a non-ceruloplasmin copper increase in serum/plasma [28,29,30,32,33], that accounts for copper excess in the bloodstream [34], in line with a number of clinical studies (Table 1).

Clinical studies also provided evidence of copper association with the severity of the disease in terms of performance in neuropsychological test batteries [38,48], disease stage [35,40,43,44] and electroencephalography (EEG) brain rhythms alterations, atrophic and cerebrovascular burden [16,49].

In the AD brain, the progressive increase of the labile copper pool is consistent with the parallel presence of an expanded pool of non-ceruloplasmin copper in the blood [38,50]. Copper disturbance in AD can be described by a loss of functional copper from protein-bound pools that reduces energy production and oxidative stress control, and a gain of redox-toxic function that is described by a bigger pool of copper loosely bound to proteins [16]. Non-ceruloplasmin copper increases the susceptibility to AD approximately threefold [42,43] and it is also associated with a higher frequency of specific variants of the *ATP7B* gene [45,46,47,51,52,53,54,55,56,57,58,59,60,61] (Table 1). The gene variants rs1061472 and rs732774 of *ATP7B* are single nucleotide polymorphisms (SNPs) that modify properties of ATPase7B protein and are associated with a higher risk of AD and with the presence of a higher fraction of non-ceruloplasmin copper in serum [38,55,58].

### 2.3. Copper Connection with Alzheimer’s Disease: Neuroscience

Several scientific articles evaluating measurements of electrical activities in cell culture models have been included in this category [17,62]. A bulk of evidence in this category comes from studies in experimental models focused on the investigation of the causative correlation of copper and non-ceruloplasmin copper in AD development and progression (Table 2) [63,64].

Huat et al. [74], in a recent review, stated that “considering the robust evidence for copper’s essential roles in the brain, it is not surprising that many studies have proposed that an imbalance in its homeostasis is associated with neurodegenerative disorders”.

In a seminal study published in 2003, Sparks and Schreurs demonstrated that in a cholesterol-fed rabbit model of AD, adding trace amounts of 0.12 mg/L copper to distilled drinking water resulted in significantly enhanced cognitive waning. It also exacerbated Aβ plaque deposition to that of control animals [65]. The study by Singh et al. provided a specific emphasis on the causative role that non-ceruloplasmin copper might play in AD onset and progression [72]. Singh et al. [72] studied normal mice (wild type) and a mouse model of AD (AβPP transgenic mice) exposed to 0.13 mg/L of copper sulfate for 90 days levels of copper via drinking water, which doubled plasma concentrations of non-ceruloplasmin copper. This fact caused either a reduction of CSF Aβ clearance across the blood brain barrier in wild-type mice or an identical effect, along with an increase in Aβ production in the transgenic mice. Thus, proposing the concept that non-ceruloplasmin copper is a causative risk factor for AD.

Thus, the relationship between copper and AD has been extensively researched in recent years.

Chronic exposure to copper and its dyshomeostasis has been linked to accelerate cognitive decline and potentially to increase the risk of AD [17,34,75]. However, copper ions due to their redox ability have been considered to be the main potential therapeutic targets in AD, and a considerable number of ligands have been developed in order to modulate the toxicity associated with copper in this context, via disruption of the Aβ-copper interaction [76].

## 3. Use of Copper in Agriculture

### 3.1. Copper Used in Plant Disease Management

Copper has been used in agriculture as a fertilizer and in the management of plant diseases. Organic agriculture is very dependent on copper as a fungicide. Several fungicides have copper in their formulation. The first fungicide to be used in all cropping systems worldwide and most famous there is the Bordeaux mixture (25% CuSO_4_). The Bordeaux mixture and, consequently, copper has been used in agriculture for more than 160 years in the management of plant diseases [77]. Fishel [78] stated that, during the 1850s in the Bordeaux region in France, a vineyard farmer was having trouble with people stealing grapes from his vines. He applied a mixture of copper and lime to part of his vineyards to make the grapes unattractive. The result was that in the plants where the copper-lime mix was applied, there was no plant disease incidence.

Nowadays copper is mostly used to control the following plant diseases: Grape downy mildew, caused by the *Plasmopara viticola*, which is a highly damaging disease for grapes, particularly in oceanic climates; Apple scab, caused by the *Venturia inaequalis*; Potato late blight, caused by the *Phytophthora infestans*, responsible for a severe disease affecting potato production. In tropical regions, there is an occurrence of the Coffee Rust Disease caused by *Hemileia vastatri* and the cocoa Witches’ Broom Disease, caused by *Crinipellis perniciosa* [79,80,81].

The Bordeaux mixture is widely used in organic agriculture worldwide since it is considered to have low toxicity for humans and the environment. Also, other fungicides containing copper in the forms of hydroxide, oxychloride, oxide, and octanoate, can be used in Organic Agriculture. However, they need authorization from the certifiers of organic products to minimize the accumulation of copper in the soil [82]. In Brazil, the recommendation follows specific legislation similar to that proposed by FAO [83,84,85].

In Europe, during the 1950s, copper in quantities of 20 to 30 kg ha^−1^ year^−1^, and sometimes even more than 80 kg ha^−1^ year^−1,^ was applied to crops to protect the plants. In Germany, between 2010 and 2015, on average, in organic farming of hops, grapes, potatoes, apples, squash, and pears copper amounting to 3.1, 2.2, 1.5, 1.5, 1.4, and 1.3 kg ha^−1^ yr^−1^, respectively, was used. In that country, the application of copper is restricted to 3 kg ha^−1^ yr^−1^ (4 kg ha^−1^ yr^−1^ for hops) [86]. In 2013, a survey investigating copper use in Germany [87] revealed that the amounts of copper used per hectare in conventional grape (0.8 kg ha^−1^), hop (1.7 kg ha^−1^), and potato-farming (0.8 kg ha^−1^) were well below those used in organic farming for the same crops (2.3, 2.6, and 1.4 kg ha^−1^, respectively).

The United States Department of Agriculture (USDA) [88] included several copper-based substances in ‘The National List of Allowed and Prohibited Substances’ in organic agriculture in the United States of America (USA). For instance, copper sulfate is as an algicide in aquatic rice systems and used as tadpole shrimp control in aquatic rice production. Its application is limited to one application per field during any 24 months, and application rates are limited to those which do not increase baseline soil test values for copper over a timeframe agreed upon by the producer and the accredited certifying agent. The USA’s legislation indicates that copper-based materials must be used in a manner that minimizes accumulation in the soil and shall not be used as herbicides.

According to Brazilian law [83,84,85] and FAO recommendation [82], the maximum amount of copper to be applied in organic agriculture is 6 kg ha^−1^ yr^−1^. According to Motta [89], in Brazil, some certifiers limit the use of the element to 3 kg ha^−1^ yr^−1^. On the other hand, in East Africa, 8 kg ha^−1^ yr^−1^ is the maximum allowed the annual copper application in areas with organic agriculture [90,91].

Formulation of Bordeaux mixture, for use in fruit trees, contains 2–10 g/L of copper sulfate and the same amount of lime (Ca(OH)_2_) diluted in water. The application is sprayed from the vegetative phase until the fruit maturation, with intervals of 10 to 15 days between applications [92]. Natural adhesive spreaders, such as sugar (10–15 g) or skim milk (200 mL), can be used for better adherence to plant leaves [89].

For the grapevine, for example, there is no fixed volume of Bordeaux mixture to be used per hectare, which can vary between 150–700 L ha^−1^. This volume varies according to several factors, such as the type of sprayer, the size of the plants, the distance between rows of plants, the climatic conditions, the disease to be controlled, and the vegetative stage of the plant [93]. Thus, considering that the smallest application volume (150 L ha^−1^) has the highest copper content (10 g/L), the highest volume (700 L ha^−1^) and the lowest content (2 g/L), a maximum of 16 applications per year can be performed in Brazil, so that the maximum annual copper dose of 6 kg ha^−1^ yr^−1^ is not exceeded, as recommended by FAO [82].

### 3.2. Copper Accumulation in Soil and Water

In general, the copper ion is very immobile in soil. Therefore, continuous copper spraying results in the accumulation of this element in the topsoil, reaching toxic levels, possibly causing plant stress, decreasing the soil microbiota biodiversity, and reducing soil fertility [94,95,96]. That is the main reason why organic farmers try to minimize copper use [86].

Sacristán and Carbó [97] studied Spanish and Australian agricultural soils cropped with lettuce. They found that soils with higher pH and higher levels of organic matter and clay result in lower copper mobility in-depth, and therefore copper accumulates in the topsoil layers. However, in general, the toxic effect of copper in plants increases as pH values decrease, due to a rise in the copper bioavailability. On the other hand, in a study conducted in an agricultural region of Haining County in southeast China, Wu et al. [98] found that the copper availability ratio and available copper concentration were decreased as a function of decreasing pH in acid soils (pH < 6.5), and increased with increasing pH in alkali soils (pH > 7.5). In Chile, Ávila et al. [99] found that copper toxicity for earthworms was lower in soils with a higher than 3.5% organic matter (OM) content, likely due to the change in bioavailable copper ions. The mitigating effect of OM was exact for soils with up to 500 mg kg^−1^ of copper.

The copper exists in soils mainly (60%) water soluble, exchangeable and sorbed forms of total copper in the upper part of soil profiles and the percentage decreases with increasing depth [100]. In Australia, total copper content in soil of Victorian vineyards is five to fifty times as high as that found in natural soils. Moreover, copper content in soil decreases with increasing distance from vines [100].

Soil copper concentration of 100 mg kg^−1^ influences rice growth, and 10% or more of grain yield, straw weight, and root weight were lost [101]. Besides copper use for plant protection, the use of pig slurry can contribute to copper accumulation in the soil. The tillage system determines the distribution of copper in the soil layers and can be used as a tool to avoid accumulation in the topsoil [102].

Copper release into water occurs through soil erosion, industrial discharge, sewage-treatment, and antifouling paints. Thus, erosion of soils that contain soil particles with adsorbed copper can result in increased copper concentrations in rivers and lakes [103]. Certainly, there is generally no contamination of the groundwater table due to copper applications in agriculture. Copper has a low mobility in the soil and its mobility decreases with increasing clay or organic matter content [97].

At present, the main concern is the increase of copper content during the distribution of drinking water, because many pipes and plumbing fixtures contain copper, which can leach into the drinking water [103]. Certainly in the last few years the use of PVC pipes instead of copper pipes has reduced the concentration of copper in the water. However, this is a problem that still persists in old houses that have copper plumbing for heated water, because this water is often used to cook food.

## 4. Agricultural Use to Copper and Its Link to Alzheimer’s Disease

Shen et al. [104] showed that the prevalence of AD in the regions in Mainland China with the highest soil copper contents (60–80 mg kg^−1^) reached 2.6 times compared to the regions that had the lowest copper contents (20–40 mg kg^−1^).

Morris et al. [105] evaluated 3718 Chicago residents with 65 years and older. They verified that high dietary intake of copper in conjunction with a diet high in saturated and trans fats might be associated with accelerated cognitive decline. Brewer [75] proposed that copper bivalent (Cu^2+^) ingestion is a major new environmental risk factor for AD. This author commented that Cu^2+^ is mainly abundant in water and food supplement pills and is much more damaging to cognition. The question remains how much and through which pathways copper used in agriculture ends up in the human diet. Copper is applied on the vegetation and only a small part is absorbed by the leaves and fruits and translocated in the plant. Copper reaches the soil where it is immobilized mainly by soil organic matter and clay particles. Plant roots take up copper as well.

The European Food Safety Authority (EFSA) report [106] makes reference to studies claiming that plants never take up more copper then the nutritional amount needed. However, this is in sharp contrast with other studies carried out in the field of phytoremediation, showing that arable and vegetable crops can take up much more copper then needed and allocate this to root, shoot and fruit tissue. By removal of the biomass, sites can be cleaned from heavy metal and copper pollution [107]. When using crops, it is relevant to know in which plant tissues copper accumulates. Tomato fruit and roots were very effective and in this case fruits should not be used for consumption [107]. Another study showed that Amaranthus, Indian Mustard and Sunflower shoot tissue were able to take up two to four times higher copper amounts than the roots, especially in fertilized soils [108]. EFSA experts [106] further conclude that insufficient scientific studies are available to conclude on the amount of copper residues in some key crops like grapes, tomatoes and cucurbits cropped following the Good Agricultural Practices guidelines. Therefore, more research is needed to clarify how much copper in organic food is bivalent (Cu^2+^).

Hummes et al. [109] evaluated the centenarian vineyard in the Pinto Bandeira municipality, the northeastern region of the state of Rio Grande do Sul, Brazil. The evaluated vines have been sprayed with copper for more than 100 years. Copper in root and leaf tissues reached 123,00 and 6800 mg kg^−1^, respectively. In grape juice and wine, copper was 9.08 and 0.78 mg L^−1^, respectively. However, copper levels in grape juice exceeded by 908% the limit established by Brazilian and international norms. On the other hand, Santos et al. [110] verified that the reduction in the copper content during the last stages of the wine cycle is related to its precipitation in the storage tanks. The copper content in the vineyard soils and, consequently, in grape must and wines is related to different factors such as the total amount of copper fungicide applied during the production period, the number of days between the last application and harvest, biological specificity of cultivars during growth, and the varietal characteristics [111].

Copper has a propensity for the accumulation in the root tissues with little upward movement towards shoots, fruits, and seeds [112]. Alexander et al. [113] verified that carrot and pea cultivars exhibited significant differences in the accumulation of copper from contaminated soil. That demonstrates that the utilization of cultivars that accumulate lower copper can yield better food.

Garrido and Botton [114] recommended a seven-day grace period (the period between the last application and the harvest) for spray products using copper in grape plants. When farmers use copper-oxychloride, the grace period is fourteen days for beans, and seven days for potato, eggplant, carrot, papaya, watermelon, peppers, and tomatoes [115]. We suggest that the grace period needs to be researched further because the copper residual contamination is dependent on the copper content in the spraying fungicide solution. Indeed, the new upper limits and recommendations of copper should be indicated in the future, due to its high impact on the soil, ecosystem, and human health.

## 5. Human Health Risk Assessment Related to Copper Use in Agriculture

The EFSA conducted a review about the pesticide risk assessment of the active substance of copper compounds [106]. That report points out the difficulties in replacing copper in plant disease management in organic agriculture. At the same time, it touches the relationship between copper and AD stating there is no scientific evidence for this, without citing any reference. However, there is evidence in numerous papers suggesting the existence of a relation between copper accumulation in the human body at aging in those individuals with a predisposition to copper metabolic abnormalities that can be identified by having higher than normal levels of non-ceruloplasmin copper and the relationship with AD development.

Massie et al. [116]; Zatta et al. [117]; Vasudevaraju et al. [118] indicated that copper accumulation is real in aging animals and the human body. However, more than the accumulation, it is the dislocation of copper as loosely bound copper also in the brain that can cause the toxicity [32]. When bound to proteins, copper is not toxic even though it increases. Conversely, when non-ceruloplasmin copper exceeds the normal reference range [116,117], it is toxic. Furthermore, recent studies showed that frequencies of the functional SNPs rs1061472 A > G and rs 732774 G > A, as well as frequency of the haplotype containing the derived GA risk alleles are associated with increased levels of non-ceruloplasmin copper and with an increased risk of AD [55] (Table 1). This evidence is in line with the EFSA statement and suggests that human carriers of *ATP7B* functional SNPs [65] exhibit higher than normal values of non-ceruloplasmin copper and are therefore prone to a copper susceptibility that can increase the risk of developing AD.

Furthermore, Zatta et al. [117] evaluated the copper contents in young (8 months) and adult (9 to 12 years) bovine brains, verifying that the copper contents varied from 1.67 to 15.7 mg g^−1^ of fresh tissue, respectively, indicating that, with aging, there is an accumulation of copper in the brain. Copper accumulation with aging has also been demonstrated to occur in humans. Vasudevaraju et al. [118] categorized human brains into three groups. Group I: below 40 years, Group II: between 41 and 60 years, and Group III: above 60 years. They found that Cu and Fe contents are significantly elevated while Zn is significantly depleted as one progresses from Group I to Group III, indicating changes in Cu and Fe with aging in the frontal cortex and hippocampus. Furthermore, those authors found that the elevation of metals was higher in the frontal cortical region compared to the hippocampal region. As a matter of fact, Vanacore et al. [119] confirmed relationships between a high content of copper and a higher incidence of AD. As previously explained, copper is a nutrient to humans and participates in the composition of important proteins [120]. However, when excess non-ceruloplasmin copper generates, it can lead to oxidative stress and other harmful effects [121].

To confirm this statement, in addition to data reported in Table 1 and Table 2, Hsu et al. [122] showed the environmental and dietary exposure to copper and its cellular mechanisms linking to AD. Moreover, Pal et al. [123] reported that autopsy of brains of AD-affected patients shows the presence of abnormally high contents of copper in the deposited Aβ plaques, while a significantly higher level of copper was found in the serum of patients suffering from Type 2 Diabetes mellitus.

Based on the evidence produced by meta-analyses, large population studies, clinical and genetic as well as experimental in vitro and animal studies that we have summarized in this review paper, we think EFSA could make a bigger effort to provide an up-to-date overview about the possible causative link between disturbed copper homeostasis and AD pathology in humans. It is clear that copper use in agriculture is not the only source of excessive copper uptake in humans, but the existing evidence about the biological connection of copper imbalance to AD together with the toxic effect of copper on soil organisms mentioned earlier in this review, has reached such a critical mass that a government action is needed. We hope to stimulate supranational discussion about this topic to invest more in research with the aim to reduce the use of copper in agriculture and the risk of developing AD among those individuals with a predisposition to copper metabolic abnormalities while safeguarding the soil biological diversity and the agroecosystem functions they provide.

## 6. Agroecological Strategies to Reduce the Use of Copper in Agriculture

In the previous section of this paper, we have given an overview of the scientific evidence sustaining that excessive use of copper in agriculture that can lead to severe soil pollution. Through the food chain, copper may accumulate in the human body, causing health problems in those people with a predisposition to copper imbalance, as those carriers of the *ATP7B* functional SNPs rs1061472 A > G and rs 732774 G > A [53].

Since 2001, the European Union and its member states have financed a wide variety of projects aimed at decreasing the reliance on copper use, especially in organic agriculture [124]. In Germany, the Federal Program for Organic Farming and Other Forms of Sustainable Agriculture (BOLN) has provided substantial funding for research projects. Research efforts focused on the further development of resistant varieties and forecast models, improved cultivation techniques and spraying techniques, the introduction of new copper products with low copper contents, improvement of copper pesticide impact assessment, development and introduction of copper-free alternatives and implementation and optimization of overall plant protection strategies [87].

In organic viticulture in northern and southern Italy, reduced and conventional dosages of traditional copper solutions were compared against copper peptidase, a new solution that has a lower than standard copper concentration thanks to its increased capacity to penetrate the pathogen cell [125]. Unfortunately, also the phytotoxicity against the crop was high. A UK study [126] concluded, based on one-year data, that apple production was higher while apple scab (*Venturia inaequalis*) infestation was lower in apple trees in an agroforestry system compared to traditional orchard systems. This result may have implications for reduced copper dosages in agroforestry systems. In tomato, the dependency on copper in crop protection against *Phytophthora infestans* could be reduced by using low copper dosages. Other research efforts have concentrated on systematic testing of natural products, among which chitosan, against a wide variety of diseases as shown in a comprehensive review by La Torre et al. [124]. Some of these natural products, such as leaf licorice extract, *Yucca schidigera Roezl ex Ortgies* extract, or potassium hydrogen carbonate [127], were effective against tomato diseases thanks to the preventive activity and direct effect on mycelial growth and sporangia germination.

Besides the simple substitution of copper by other natural or synthetic products [125,127] or new copper solutions with similar efficacy but lower copper concentrations, agroecological approaches for the design of sustainable cropping systems need to be taken into consideration. In the first place, agroecological approaches would guarantee a detailed analysis of the cropping systems that depend on copper and take into account the specific socio-economical situation of the regional context. It would require a participatory approach involving farmers, local stakeholders, and the entire food system to determine potential solutions for a reduction or abandonment of copper applications. Agroecological solutions will be sought mainly in terms of prevention of the key disease problems in the specific cropping system. These solutions can be agronomic, but also the implementation of biodiversity-based solutions should be explored in more detail. The apple-agroforestry case study mentioned before is a good example of such an approach. The enhancement of functional biodiversity in the cropping systems aimed at the prevention and control of these key diseases can follow the approach proposed by Moonen and Bàrberi [128]. This approach foresees in the first place an analysis, preferably in a participatory-action research context, of strengths and weaknesses of the cropping system and its socio-economical context concerning the specific problem (in this case, the need to reduce reliance on copper). This first analysis will result in the definition of prioritized ecosystem services and trade-offs to be avoided (in this case, the agroecosystem services needed are disease control without copper). Experts (in this case, mainly plant pathologists and agronomists) need to identify one or more functional groups, defined as ‘clusters of elements (at gene, species, or habitat level) that provide the same agroecosystem service’ [128] that can provide the desired agroecosystem services while maintaining crop production. Based on this knowledge, the cropping system can be redesigned through participatory action research.

In support to this biodiversity-based redesign of the cropping system aimed at disease control, farmers need to consider agronomical solutions to reduce the dependency of plant protection products (PPPs) by optimization and increased use of preventive measures like cultural measures and non-chemical plant protection as suggested by IFOAM EU GROUP [86]. For example, in annual and horticultural cropping systems, crop rotations are effective in disease control, as shown by Bankina et al. [129]. They reported a significant decrease in wheat stem rot disease in diversified crop rotations. Also, the use of cover crops can be considered to diversify crop rotation and break diseases. However, disease control is effective in the early crop stage but does not last until crop harvest, as shown by Runno-Paurson et al. [130] for potato blight control in organic agriculture. Diversified crop rotations are especially relevant for disease control in conservation agriculture or reduced tillage systems since these systems are more susceptible to disease development, such as Fusarium blight in winter cereals [131]. Other agronomic interventions that can greatly affect disease development are optimization of irrigation systems as shown by Lage et al. [132] controlling powdery mildew in organic tomato production, and the selection and development of resistant or robust varieties [124]. Not much research was performed on herbaceous vegetation management for disease control in orchards and vineyards, but some studies showed that the choice of vegetation type is relevant. For example, some vegetation types were able to reduce *Phytophthora* crown or root rot in apple orchards [133]. At the same time, exotic grasses improved black foot pathogens such as *Ilyonectria liriodendri* in vineyards through alteration of the root-associated fungal communities in vineyards [134]. Also, optimization of air movement in the crop canopy through correct pruning strategies are effective [135] or in high-value crops netting can be used to optimize the microclimate, protect the crop from late frost and hail damage and therefore reduce copper-based treatments in organic agriculture [124].

Despite the evidence showing that copper causes severe damage to agroecosystems and human health and despite the effort made already to show alternatives for copper use in organic farming, in Europe, where the Current Authorization of Copper-based Antimicrobial Compounds [136] expired on 31 January 2019, no decision has been made yet [137].

Lamichhane et al. [77] commented that the formulation of appropriate policies aimed at the reduction of the risks associated with copper represents a key challenge for researchers, policymakers, and farmers soon.

## 7. Conclusions

Copper is mostly used to control plant diseases. The Bordeaux mixture (copper sulfate) is widely used in organic agriculture worldwide since it is considered to have low toxicity for humans and the environment. However, when in excess, copper can lead to oxidative stress and other harmful effects in humans with a predisposition to copper imbalance. Chronic exposure to copper in people susceptible to copper dyshomeostasis, as revealed by higher than normal levels of non-ceruloplasmin copper, has been linked to accelerated cognitive decline and an increased risk of AD.

Besides the adoption of strategies for the minimization of copper in organic farming such as the use of natural alternatives, lower application rates and the reduction of the dependency of plant protection products by optimization and increased use of preventive measures like cultural measures and non-chemical plant protection, the application of agroecological approaches for the design of sustainable cropping systems needs to be taken into consideration.

## Figures and Tables

**Figure 1 biomolecules-10-00897-f001:**
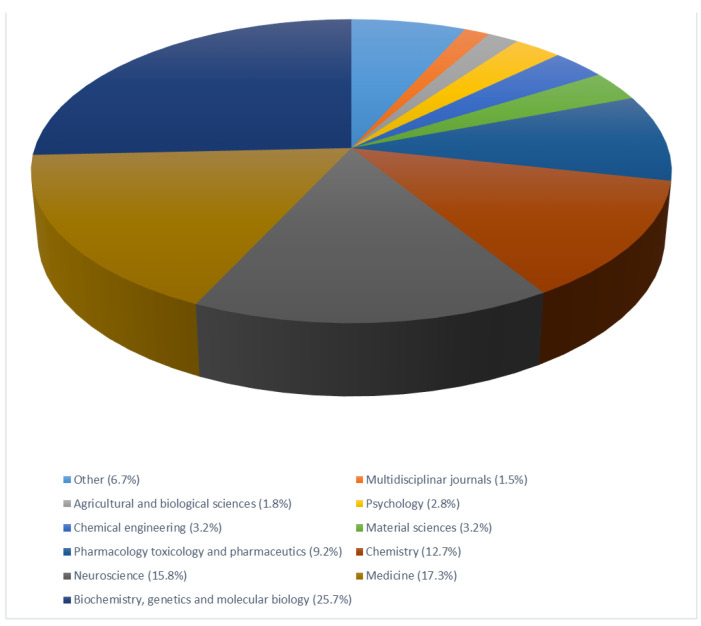
Pie chart illustrating the literature published on the topic ‘Copper’ and “Alzheimer’s disease”. Search on the Scopus research engine retrieved 3009 scientific articles; the stratification by ‘subject area’ reveals that 25.7% of the papers published come from the Biochemistry area, 17.3% from Medicine, 15.8% from Neuroscience, 12.7% from Chemistry and 9.2% from Pharmacology.

**Table 1 biomolecules-10-00897-t001:** Clinical studies analyzing the copper and *ATP7B* link to Alzheimer’s disease risk and the association with the subjects’ clinical status.

	Subjects	Risk (OR ^1^, RR ^2^, HR ^3^)	CI ^4^ 95%	*p* Value
**Serum Copper and Risk for Alzheimer’s Disease**				
Copper level was higher in subjects with AD than in control subjects and correlated with poor neuropsychological performance and medial temporal lobe atrophy [35]	76 AD vs. 79 healthy subjects	1.8	1.36–2.43	*p* < 0.05
Copper level was higher in subjects with AD than in patients with vascular dementia subjects (VaD) [36]	48 AD vs. 20 VaD	2.06	1.28–3.31	*p* < 0.003
Non-ceruloplasmin copper was higher in AD than in healthy controls and VaD and correlated with poor neuropsychological performance [37]	47 AD vs. 44 healthy subjects and 24 VaD subjects			*p* < 0.001
Non-ceruloplasmin copper was higher in AD than in healthy controls; Cerebrospinal (CSF) β-amyloid and H-Tau correlated with serum non-ceruloplasmin copper; copper in the CSF was partially dependent on the serum Non-ceruloplasmin copper (t = 2.2, *p* = 0.04). Mini-Mental State Examination (MMSE) and verbal memory scores correlated positively with β-amyloid (r = 0.46, *p* = 0.002) and inversely with nonceruloplasmin-Cu (= 0.45, *p* = 0.003) [38]	28 AD vs. 25 healthy subjects			*p* < 0.001
Non-ceruloplasmin copper predicted the annual change in MMSE; when the annual change in MMSE was divided into <3 or ≥3 points, Non-ceruloplasmin copper was the only predictor of a more severe decline [39]	81 AD subjects, 1 year longitudinal study	1.23	1.03–1.47	*p* < 0.022
Non-ceruloplasmin copper was higher in MCI than in healthy subjects [40]	83 MCI subjects, 100 healthy subjects	1.22	1.05–1.41	*p* < 0.01
Copper level showed a significant increase in the serum of AD and MCI compared to control (*p* = 0.038) [41]	36 AD, 18 MCI vs. 25 healthy subjects			*p* < 0.05
Non-ceruloplasmin copper increased the risk of having AD; when combined in an algorithm with sex, APOE, Cp/Tf, TAS, the ability to discriminate AD patients vs. controls was high (ROC ^5^, AUC ^6^ = 0.9) [42]	93 AD, 45 VaD, 48 healthy subjects	3.21	1.53–6.71	*p* < 0.002
Non-ceruloplasmin copper was a predictor of conversion to AD: MCI subjects with nonceruloplasmin-Cu levels > 1.6 µmol/L had a hazard conversion rate (50% conversion in 4 years) that was ~3 higher than those with values ≤ 1.6 µmol/L (< 20% in 4 years) [43]	131 MCI subjects, 6 years longitudinal study	3.3	1.21–9.24	*p* = 0.02
Non-ceruloplasmin copper levels higher in MCI and AD with respect to control (*p* < 0.0001) [44]	44 AD and 36 MCI vs. 28 healthy subjects			*p* < 0.001
Non-ceruloplasmin copper and Cu:Cp resulted higher in AD and in Wilson disease (WD) than in healthy controls; while nCp-Cu was similar between AD and WD, Cu:Cp was higher in WD. 24 h urinary copper excretion in AD patients (12.05 μg/day) was higher than in healthy controls (4.82 μg/day); 77.8% of the AD patients under D-penicillamine treatment had a 24 h urinary excretion higher than 200 μg/day, suggestive of a failure of copper control [34]	385 AD, 9 WD, 336 healthy subjects			*p* < 0.0001
Non-ceruloplasmin copper does not change in frontotemporal lobar degeneration (FTLD) [36]	85 FTLD, 55 healthy subjects			*p* < 0.001
**ATP7B Gene Variants and Risk for Alzheimer’s Disease**				
Specific genetic variants in the *ATP7B* gene, namely rs1801243 (OR = 1.52, 95% CI = 1.10–2.09), rs2147363 (OR = 1.58, 95% CI = 1.11–2.25), rs1061472 (OR = 1.73, 95% CI = 1.23–2.43), and rs732774 (OR = 2.31, 95% CI = 1.41–3.77) increased the risk of having AD [45]	285 AD vs. 230 healthy subjects	2.3	1.41–3.77	*p* < 0.001
Wilson disease-causing variant rs7334118 in linkage disequilibrium with the intronic rs2147363 (associated with AD risk) was detected in two AD patients but in no healthy individuals. However, this Wilson disease mutation did not explain the observed genetic association of rs2147363. Conversely, in silico analyses of rs2147363 functionality highlighted that this variant is located in a binding site of a transcription factor and is associated with regulatory functions [46]	286 AD vs. 283 healthy subjects	1.3	1.06–1.69	*p* = 0.015
Haplotype TGC in specific genetic variants in the *ATP7B* gene, namely rs1801243, rs1801249, rs1801244, and rs1801243 was associated with an increased risk of having AD [47]	120 AD vs. 111 healthy subjects	5.16	2.54–10.5	*p* < 0.001

^1^ OR, odds ratio; ^2^ RR, relative risk; ^3^ HR, hazard ratio; ^4^ CI, Coefficient Interval; ^5^ ROC, receiver operating characteristic curves depict the likelihood of a given test to be excellent, good, or worthless. The accuracy of the test depends, in this case, on the probability that a subject randomly selected from AD group has a nonceruloplasmin copper value higher than that of a subject randomly selected from the healthy control group. The accuracy is measured by the ^6^ AUC (area under the curve) of the ROC curve. An AUC = 1 represents a perfect test, a value of 0.5 represents a worthless test, and values in the 0.7–0.8 range are considered to be fair.

**Table 2 biomolecules-10-00897-t002:** Experimental models focused on the investigation of the causative correlation of copper and non-ceruloplasmin copper in Alzheimer’s disease (AD) development and progression.

Authors, Year	Animal Model	Dose and Route	Duration	Effects
**Animal Models of Copper Neurotoxicity Induced by Altered Diet**				
Sparks and Schreurs, 2003 [65]	New Zealand white rabbits	12 mg/L copper in DW ^1^ + 2% cholesterol − oral	10 weeks	Accumulation of Aβ in brain; deficit in complex memory acquisition
Sparks et al., 2006 [66]	Beagle dogs	200 mg/L CuSO_4_ in DW + high fat diet − oral	4 months	Extracellular Aβ deposits
Lu et al., 2006 [67]	Kumming strain mice	0.21 mg/L copper in DW + 2% cholesterol − oral	8 weeks	Cognitive deficits; neuronal apoptosis
Arnal et al., 2013 [68]	Wistar rat	3 mg/L copper in tap water + 2% cholesterol − oral	2 months	Increased oxidative stress in brain; increased non-ceruloplasmin copper in hippocampus; increased Aβ (1–42)/ Aβ (1–40) in cortex and hippocampus
Arnal et al., 2013 [68]	Wistar rat	3 mg/L copper in tap water + 2% cholesterol − oral	8 weeks	Slight nut noticeable change in visuo-spatial memory
Yao et al., 2018 [69]	Tg2567 mouse	0.1 mg/L copper in drinking water and 2% cholesterol in the food	3 months	Significant deposit of Aβ and senile-plaque formation in hippocampus and temporal cortex regions
Abolaji et al., 2020 [70]	D. melanogaster flies	Cu^2+^ (1 mM)	7 days	reduced survival
Lamtai et al., 2020 [71]	Rat	CuCl_2_ (0.25 mg/kg, 0.5 mg/kg and 1 mg/kg) injected intraperitoneally	8 weeks	Working memory, spatial learning and memory were significantly impaired in rats treated with Cu at dose of 1 mg/kg
**Models of Copper Neurotoxicity in Genetically Compromised Animals**				
Sparks et al., 2006 [66]	Watanable rabbits	0.13 mg/L copper in DW − oral	10 weeks	Accumulation of Aβ in superior temporal cortex and hippocampus
Sparks et al., 2006 [66]	PS1/APP transgenic mice	0.12 mg/L Cu in DW − oral	6 weeks	Deposition of Aβ
Singh et al., 2013 [72]	APP sw/0 mice	0.13 mg/L copper in DW − oral	90 days	Increase brain Aβ production; increased neuroinflammation; memory impairment; increased Cu levels in brain capillaries and parenchyma
Yu et al., 2014 [73]	3xTg-AD	250 mg/L CuSO_4_ in drinking water	6 months	memory impairment

^1^ DW, deionized water.

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
