# Peer review of "Agricultural Use of Copper and Its Link to Alzheimer’s Disease"

_biomolecules, 2020, doi:10.3390/biom10060897_

Round 1
Reviewer 1 Report
The review is an interesting summary of our actual knowledge about copper use in agriculture and its possible consequences on human health. Differences concerning copper uses and concentrations between countries are described, as well as mechanisms of action. In particular, the link between cognitive decline and an increased risk of Alzheimer’s disease development are highlighted.
Future directions and perspectives are discussed, which adds to the comprehensive references that are listed.
The reading is fluent, although some improvements can be done.
In particular:
-I would suggest revising English.
e.g. Line 41: Alzheimer’s disease (AD) is the most cause of dementia in the elderly
-When talking about oligomannate, I suggest briefly explaining the supposed mechanism of action.
-fig1: image is too big if compared to the color legend. Number of total articles could appear in the figure, in order to be more direct in communicating the result to the reader.
-line 111 “most of these studies”. Better: most of the cited studies…or: most of the studies currently available in literature…
Line 120: “and exchanged among them”. This is not clear.
-in table 2, specify the way of administration in D. melanogaster flies.
Also, “in drinking water” is an oral way of administration. Specify the difference with “oral” in the table.
-line 180: linked to “accelerated” cognitive decline or: suggested to accelerate cognitive decline
-line 300: in sharp contract? (contrast?)
To conclude, I think this is a comprehensive review about a hot topic that researchers need to further investigate. Partial text revision and a refining of images can easily increase its impact on the scientific community.
Author Response
Reviewer 1
Comments and Suggestions for Authors
The review is an interesting summary of our actual knowledge about copper use in agriculture and its possible consequences on human health. Differences concerning copper uses and concentrations between countries are described, as well as mechanisms of action. In particular, the link between cognitive decline and an increased risk of Alzheimer’s disease development are highlighted.
Response: thanks for the comment, please find the revised text as yellow highlighted in the new version of the manuscript
Future directions and perspectives are discussed, which adds to the comprehensive references that are listed.
The reading is fluent, although some improvements can be done.
In particular:
Point 1: -I would suggest revising English.
e.g. Line 41: Alzheimer’s disease (AD) is the most cause of dementia in the elderly
Response: the sentence has been emended in Alzheimer’s disease (AD) is the most common cause of dementia in the elderly (p.1,l.41)
Point 2:. -When talking about oligomannate, I suggest briefly explaining the supposed mechanism of action.
Response: The development of Oligomannate was inspired by the relatively low occurrence of Alzheimer’s disease among elderly people who regularly consumed seaweed. The Dr Geng Meiyu research team (Chinese Academy of Sciences’ Shanghai Institute of Materia Medica), started looking into possible connections and, in 1997, identified a unique sugar in seaweed which might play an important role in the phenomenon. In September 2019, the authors published a paper that they had found that Oligomannate can reduces the formation of a protein harmful to neurons and regulate the microbiota in human intestines to reduce the risk of brain inflammation. Some of these concepts are now reported in the new version of the manuscript (p. 2, ll.17-20; Ref [8]).
Point 3: -fig1: image is too big if compared to the color legend. Number of total articles could appear in the figure, in order to be more direct in communicating the result to the reader.
Response: a new Figure 1 has been included in the revised version of the manuscript according to reviewer’s suggestion
Point 4: -line 111 “most of these studies”. Better: most of the cited studies…or: most of the studies currently available in literature…
Response: we modified the sentence as suggested
Point 5: Line 120: “and exchanged among them”. This is not clear.
Response: the sentence has been corrected as follow: ‘and it is an exchangeable copper pool’
Point 6: -in table 2, specify the way of administration in D. melanogaster flies.
Response: " added into the medium - oral" has been included
Point 7: Also, “in drinking water” is an oral way of administration. Specify the difference with “oral” in the table.
Response: the sentence has been corrected as follow: ‘in drinking water - oral’
Point 8: -line 180: linked to “accelerated” cognitive decline or: suggested to accelerate cognitive decline.
Response: we modified the sentence as suggested
Point 9: -line 300: in sharp contract? (contrast?)
Response: we modified the sentence as suggested – “contrast”
Point 10: To conclude, I think this is a comprehensive review about a hot topic that researchers need to further investigate. Partial text revision and a refining of images can easily increase its impact on the scientific community.
Response: we modified the review as suggested

Reviewer 2 Report
-
In this manuscript Coelho and coauthors., nicely reviewed the agriculture use of Copper and its link to the risk for Alzheimer's Disease. Specifically they summerize evidences reporting that copper can lead to oxidative stress and may have other harmful effects in humans with a predisposition to copper imbalance. Chronic exposure to copper in people susceptible to copper dyshomeostasis, as revealed by higher than normal levels of non-ceruloplasmin copper, has been linked to accelerated cognitive decline and an increased risk of AD. The review is well presented and organized, fluent and clearly written. My only suggestion is to combine paragraphs 4 and 5, since they are both mainly focused on AD risk. Alternatively, paragraph 5 should be expanded to other diseases’ risk if present.
Author Response
Point 1: My only suggestion is to combine paragraphs 4 and 5, since they are both mainly focused on AD risk. Alternatively, paragraph 5 should be expanded to other diseases’ risk if present.
Response: we agree with the reviewer. Following the reviewer's suggestion, we have combined the two sections (4 and 5) that have now the title of: “4.Agricultural use to Copper and its link to Alzheimer's Disease”.

Reviewer 3 Report
Excellent review in a very hot and interesting topic. Please accept it as it is for publication
Author Response
Thanks for the comment